# Collider bias and the apparent protective effect of glucose-6-phosphate dehydrogenase deficiency on cerebral malaria

James A Watson[1,2†]*, Stije J Leopold[1,2†]*, Julie A Simpson[3], Nicholas PJ Day[1,2], Arjen M Dondorp[2], Nicholas J White[2]*

[1]Mahidol Oxford Tropical Medicine Research Unit, Faculty of Tropical Medicine, Mahidol University, Bangkok, Thailand; [2]Nuffield Department of Medicine, Centre for Tropical Medicine and Global Health, University of Oxford, Oxford, United Kingdom; [3]Centre for Epidemiology and Biostatistics, Melbourne School of Population and Global Health, The University of Melbourne, Melbourne, Australia

**Abstract** Case fatality rates in severe falciparum malaria depend on the pattern and degree of vital organ dysfunction. Recent large-scale case-control analyses of pooled severe malaria data reported that glucose-6-phosphate dehydrogenase deficiency (G6PDd) was protective against cerebral malaria but increased the risk of severe malarial anaemia. A novel formulation of the balancing selection hypothesis was proposed as an explanation for these findings, whereby the selective advantage is driven by the competing risks of death from cerebral malaria and death from severe malarial anaemia. We re-analysed these claims using causal diagrams and showed that they are subject to collider bias. A simulation based sensitivity analysis, varying the strength of the known effect of G6PDd on anaemia, showed that this bias is sufficient to explain all of the observed association. Future genetic epidemiology studies in severe malaria would benefit from the use of causal reasoning.
DOI: https://doi.org/10.7554/eLife.43154.001

*For correspondence:
jwatowatson@gmail.com (JAW);
stije@tropmedres.ac (SJL);
nickw@tropmedres.ac (NJW)

†These authors contributed equally to this work

Competing interests: The authors declare that no competing interests exist.

## Introduction

Severe falciparum malaria is defined by one or more criteria indicating vital organ dysfunction in the presence of microscopy confirmed asexual blood stages of *Plasmodium falciparum* in the peripheral blood film (**WHO, 2015**). Multiple vital organ dysfunction is associated with increased mortality (**WHO, 2014**). Common major clinical manifestations of severe malaria include coma, acidosis, renal failure and anaemia. Of these manifestations, anaemia is an inevitable consequence of symptomatic malaria (**White et al., 2014**). However, anaemia in individuals at risk of *Plasmodium falciparum* infection can also be the consequence of red cell genetic polymorphisms frequent in the populations at risk, such as glucose-6-phosphate dehydrogenase deficiency (G6PDd) or haemoglobinopathies.

There is considerable interest in understanding the mechanisms conferring protective effects against severe falciparum malaria of the genetic polymorphisms which are common in malaria endemic areas (**Weatherall, 2008**). For some, such as the sickle cell trait, several different mechanisms have been proposed. These include reduced parasite erythrocyte invasion, enhanced parasitised red cell phagocytosis and a reduced propensity of infected red cells to sequester in the microvasculature (**Malaria Genomic Epidemiology Network et al., 2014**; **Band et al., 2015**; **Cholera et al., 2008**; **Williams, 2016**). The mechanism underlying protection from severe falciparum malaria is less clear for others such as glucose-6-phosphate dehydrogenase deficiency (G6PDd). This

X-linked genetic polymorphism results in the most common human enzymopathy. Nearly 200 different genetic variants have been reported (*Howes et al., 2012*; *Luzzatto and Arese, 2018*). The mechanism whereby G6PD deficiency protects against malaria, and the natural selection forces which have resulted in the different genotypes are still debated. Prospective observational hospital or clinic based patient studies have provided the major component of the evidence base. Estimating causal effects from observational studies in severe malaria patients is difficult due to both confounding and selection bias. This work focuses on collider bias introduced by inappropriate data filtering (*Snoep et al., 2014*; *Pearce and Richiardi, 2014*).

It has been suggested that G6PDd both increases the risk of severe malarial anaemia (SMA) and decreases the risk of cerebral malaria (CM) (*Malaria Genomic Epidemiology Network et al., 2014*; *Clarke et al., 2017*). These conclusions were based on a pooled analysis of observational data from over 11,000 patients with severe malaria studied in Africa and Asia, and relevant population controls. Based on these genetic association studies, a new formulation of the balancing-selection hypothesis was proposed in which G6PD polymorphisms are maintained in human populations, at least in part, by an evolutionary trade-off between different adverse outcomes of *P. falciparum* infection (*Clarke et al., 2017*). Collider bias probably explains this negative association between G6PDd and CM, suggesting that causal interpretations of this association and the novel formulation of balancing selection in G6PDd are invalid.

## Results

Two published analyses of pooled data from observational studies of patients with severe falciparum malaria used severe malarial anaemia (SMA) and cerebral malaria (CM) as the main endpoints (outcomes) of interest (*Malaria Genomic Epidemiology Network et al., 2014*; *Clarke et al., 2017*). Both these published analyses defined cases of CM as the presence of coma but without concomitant SMA, and cases of SMA as patients with severe anaemia but who were conscious. Therefore, these case definitions excluded patients who had both SMA and CM. All other presentations of severe malaria were also excluded (pulmonary oedema, shock, etc.). Population controls were recruited at each site to match the ethnic composition of cases, and in some instances cord blood samples were used as controls. The consequence of these case definitions is to create an artificial dependency between SMA and CM: if a patient has SMA then they cannot have CM. G6PDd is known to influence haemoglobin concentrations directly by causing haemolysis of older erythrocytes in acute malaria. Therefore it is to be expected that SMA is positively correlated with G6PDd, thus creating a negative correlation between CM and G6PDd. In probabilistic terms, this conditional dependence is written as $P(\text{SMA}|\text{CM}) \neq P(\text{SMA})$. Indeed, when all the G6PDd mutations were mapped onto the WHO severity classification score (*Yoshida et al., 1971*), it was observed that "The mean G6PDd score was 13.5% in controls, 13% in cerebral malaria cases and 16.9% in severe malarial anaemia cases [..]." (page 8, *Clarke et al., 2017*). This pattern remained consistent (6, 5.6, and 7.1%, respectively) after exclusion of the G6PD c.202C > T mutation (one of the 'A-' mutations and the most prevalent in the pooled data).

By excluding patients with both SMA and CM (approximately 12% of those with either SMA or CM in the pooled data), the number of G6PDd patients in the CM category is artificially reduced and we would expect there to be fewer G6PDd patients than in the control group. *Figure 1* proposes a simple causal diagram which posits plausible inter-dependencies limited to the variables of interest. A simple simulation study based on the assumptions shown in *Figure 1* can be used to estimate the relationship between G6PDd and SMA which would result in the observed odds ratio for G6PDd in CM cases versus controls reported in *Clarke et al. (2017)*. We assessed the null model in which there is no direct causal link between G6PDd and CM in severe falciparum malaria (i.e. no arrow from G6PD deficiency to CM). We also assume that there is no direct link between SMA and CM. We calibrated the model with the marginal probabilities of SMA and CM reported in *Clarke et al. (2017)* and only varied the odds ratio of G6PDd in SMA cases versus controls from 1 (assuming no effect of G6PDd on SMA) to 2 (twice as likely to be G6PDd in the SMA cases than in the controls). For simplicity (avoiding assumptions concerning gene dose effects) we restricted the analysis to males and only compare the simulation results with the reported associations in males.

*Figure 2* shows that if the odds ratio for G6PDd in SMA cases versus controls is strictly greater than 1, the estimated odds ratio for G6PDd in CM cases versus controls is biased (the thick red line

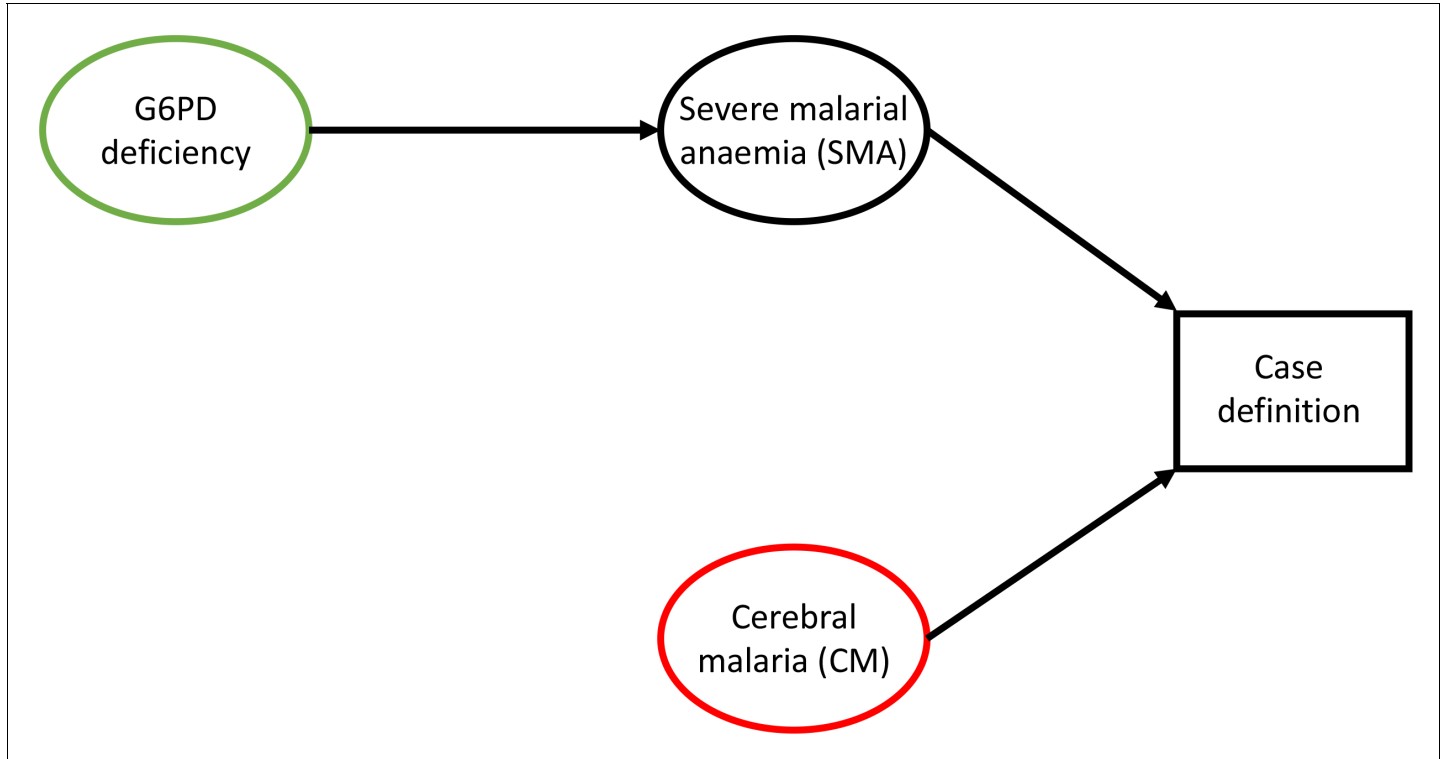

**Figure 1.** Causal diagram highlighting collider bias in *Clarke et al. (2017)* and *Malaria Genomic Epidemiology Network et al. (2014)*. G6PD deficiency is the exposure of interest (green) and cerebral malaria (CM) is the outcome of interest (red). By defining the CM cases as those who had coma but no severe anaemia, collider bias operates on the effect of G6PDd on CM.
DOI: https://doi.org/10.7554/eLife.43154.002

is below the true simulated value of 1). The magnitude of this bias increases monotonically as the odds ratio for G6PDd in SMA cases versus controls increases. As can be seen from the causal diagram in *Figure 1*, there is no bias in the estimated odds ratio for G6PDd in SMA cases versus controls (in *Figure 2* the thick blue line approximates the identity line). This simple simulation model, restricted to males, shows that for any value of the odds ratio for G6PDd in SMA cases versus controls taken inside the reported 95% confidence interval (CI) [1.2–1.8] from *Clarke et al. (2017)*, will result in a biased odds ratio for G6PDd in CM cases versus controls inside the interval [0.69–0.98], the reported 95% CI for G6PDd in CM cases versus controls (*Clarke et al., 2017*). Moreover, if we use their reported point estimate of 1.48, restricted to males, for the odds ratio of G6PDd in SMA cases versus controls, the simulation model estimates that the observed (biased) odds ratio for G6PDd in CM cases versus controls is 0.87, qualitatively very close to their estimate of 0.82. We note that the effect of G6PDd on severe anaemia in homozygous G6PDd girls and hemizygous G6PDd boys reported in *Uyoga et al. (2015)*, an odds ratio of 1.71 (95% CI: 1.34–2.18) for G6PDd in SMA cases versus controls, is also consistent with these results. Therefore collider bias could be sufficient to explain all the observed association.

## Discussion

This re-analysis of recent reports that G6PDd reduced the risk of CM directly (*Malaria Genomic Epidemiology Network et al., 2014*; *Clarke et al., 2017*) suggests that the observations could have resulted entirely from collider bias. This highlights the difficulty of inferring causal relationships between baseline patient covariates (in this case G6PDd) and covariates which define inclusion criteria. The necessary causal odds ratios for G6PDd in SMA cases versus controls which would give rise to the biased observed association between G6PD status and CM fit with the recent estimate of 1.71 in homozygous girls and hemizygous boys (*Uyoga et al., 2015*). This is not to say that the risk

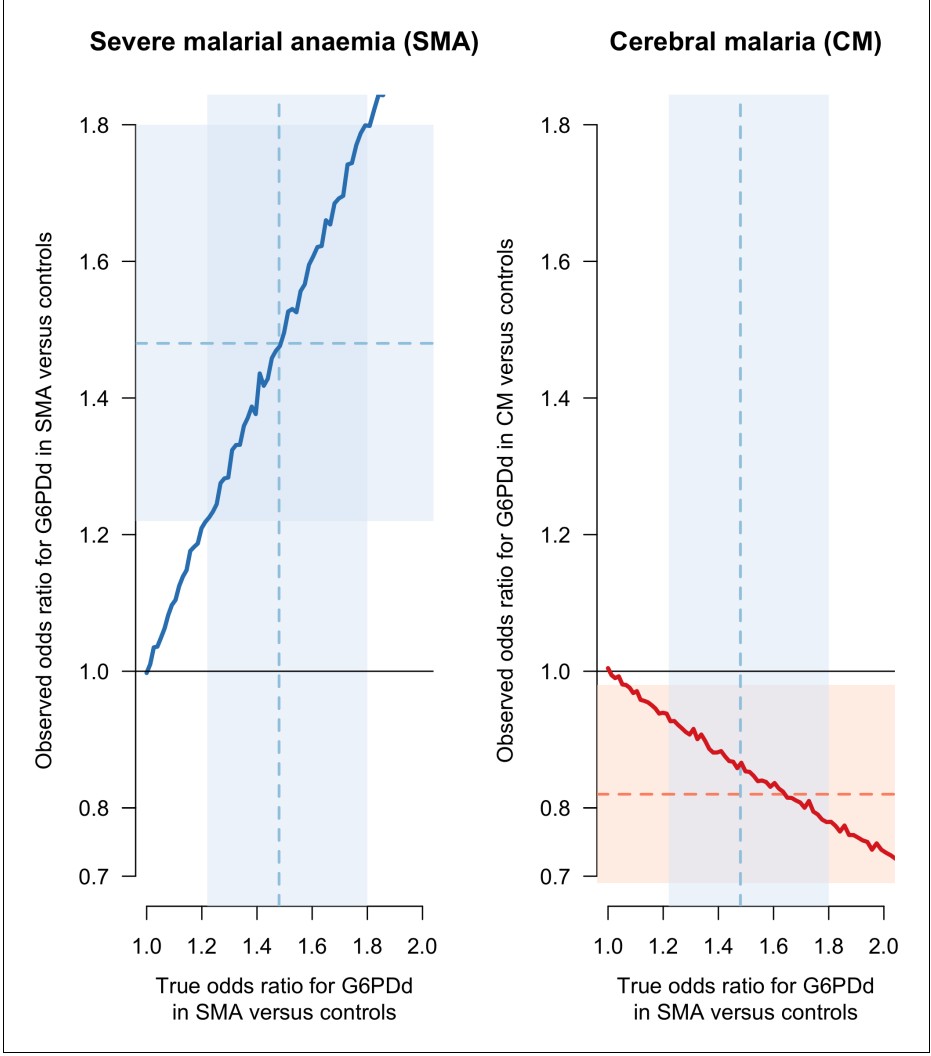

**Figure 2.** Results of the simulation based sensitivity analysis showing how collider bias can explain all the reported association between CM and G6PDd. The simulation assumes that CM is independent of G6PD status but that SMA is dependent on G6PDd status (*Figure 1*). Case definitions of CM and SMA exclude patients with both. The left panel shows the observed simulation based estimate of the odds ratio (OR) for G6PDd in SMA cases versus controls (y-axis) as a function of the true simulated value (x-axis). No bias arises (the observed and true values lie on the line of identity). The right panel shows the observed simulation based estimate of the OR for G6PDd in CM cases versus controls (y-axis), again as a function of the true simulated value of the OR for G6PDd in SMA cases versus controls (x-axis). This estimate suffers from collider bias since the true value of the OR for G6PDd in SMA cases versus controls was set to 1). The faint blue shaded areas show the 95% CI [1.22–1.8] for the odds ratio of G6PDd in SMA cases versus controls, restricted to males (*Clarke et al., 2017*). The point estimate (1.48) is shown by the dashed blue line. The faint red shaded area shows the 95% CI [0.69–0.98] for G6PDd in CM cases versus controls, also restricted to males, with the point estimate (0.82) shown by the dashed red line. CI: confidence interval.

DOI: https://doi.org/10.7554/eLife.43154.003

of CM is unaffected by G6PDd, but that the observations reported could have arisen purely as a result of the implicit collider bias induced by the selection of patients by the severe malaria criteria.

The example reported here highlights a major difficulty when attempting to estimate causal contributions when factors of interest define inclusion into the clinical study and the subsequent data analysis. All prospective observational severe malaria studies suffer from two major issues. First, case definitions are subjective and change over time (even though standard guidelines exist, see *WHO (2015)*) and mortality is strongly dependent on the case definition. Second, enrolment into

studies can only be done at the hospital or clinic level and neither duration of illness nor treatment seeking behaviour can be accounted for adequately.

The notion and definition of 'severe malaria' has two operational purposes. First it is a clinical tool for appropriate triage of malaria patients at high risk of death. Second it is a research tool for the evaluation of novel interventions seeking to reduce mortality. Interventions aimed at reducing mortality need to be trialled in the most severely ill patients in order to demonstrate intervention efficacy in this important subgroup. Pooled analyses of severe malaria studies need to take into account the variability of study inclusion and exclusion criteria. Researchers must appreciate that severe malaria is not an objective category but a subjective case definition subset from a spectrum of severity. Moreover, restricting analyses to specific patient subgroups, especially in the analysis of large pooled datasets, can have a considerable impact on the final result (*Gelman and Loken, 2013*). With increased emphasis on providing open access data so that analyses can be evaluated and best use made of clinical research it would be very helpful if investigators could publish reproducible code alongside their analyses. Future genetic epidemiological studies could benefit from use of causal diagrams and would be more readily evaluable by provision of accompanying code.

## Materials and methods

### Data analysis in *Clarke et al. (2017)* and *Malaria Genomic Epidemiology Network et al. (2014)*

The odds ratios for G6PDd in cases versus controls are given in Table 3 of *Malaria Genomic Epidemiology Network et al. (2014)* (page 1201). The results, restricted to males, are 0.81 (95% CI: 0.68–0.96) for CM and 1.49 (95% CI: 1.24–1.79) for SMA. The case phenotype definitions are denoted 'Cerebral malaria only' for CM and 'Severe malarial anaemia' only for SMA. These case definitions, whereby patients with both SMA and CM are excluded, are also given in their Table 1. A total of 6283 cases had cerebral malaria or severe malarial anaemia, broken down as 3345 had cerebral malaria only; 2196 had severe malarial anaemia only; 742 had both cerebral malaria and severe malarial anaemia. The reported odds ratios were computed using logistic regression models with the main adjustment of interest being sickle haemoglobin genotype (HbS). We only consider the reported results restricted to males. The relevant section from the paper is: "*Single-SNP tests, adjusted for HbS genotype, sex and ancestry, for association with severe malaria and the severe malaria subtypes cerebral malaria only and severe malarial anemia only were performed for the 55 SNPs with a known association with severe malaria. Standard logistic regression models were used for tests of association at each autosomal SNP (Supplementary Table 25). Primary analyses comprised tests of association between each SNP and severe malaria phenotypes across all individuals combined as well as separately by sex (X-chromosome SNPs only) and study site: genotypic, additive, dominant, recessive and heterozygote advantage genetic models of inheritance were considered.*" (online Methods, Statistical analysis, (*Malaria Genomic Epidemiology Network et al., 2014*)).

### *Clarke et al., 2017*

The odds ratios for G6PDd in cases versus controls are given in Table 3 of *Clarke et al. (2017)* (page 6). In this publication, the results, restricted to males, are 0.82 (95% CI: 0.69–0.98) for CM and 1.48 (95% CI: 1.22–1.8) for SMA (their Table 3). The reason for the slight discrepancy between the two publications does not appear to be stated. They included a total of 6284 patients with cerebral malaria or severe malarial anaemia, broken down as: 3359 individuals had cerebral malaria only; 2184 had severe malarial anaemia only; 741 had both cerebral malaria and severe malarial anaemia. Table 1 and 6 of *Clarke et al. (2017)* show the case definitions for CM and SMA, highlighting that those who have both clinical presentations are excluded from the respective case definitions. This is further confirmed on page 18 where the authors state: "*For reasons of sample size, we did not conduct a detailed analysis of other sub-types of severe malaria, or of those individuals who had both cerebral malaria and severe malarial anaemia'. Standard logistic regression models were also used to obtain the odds ratios: 'In primary analyses, standard fixed effects logistic regression methods were used for tests of association with severe malaria and sub-types at each SNP under additive,*

*dominant, recessive and heterozygous models. [..] Results were adjusted for sickle hemoglobin (HbS), gender and ethnicity."* (bottom of page 18).

## Sensitivity analysis

In order to characterise the proportion of the reported association explained by collider bias we constructed a simple simulation study based on the analytical procedures in *Malaria Genomic Epidemiology Network et al. (2014)*; *Clarke et al. (2017)*. We restrict our simulation to males, and calibrate and test the model using only the results reported in males in both publications (identical up to one decimal point). For simplicity, the simulation ignores the effect of HbS which is a confounder between SMA and CM, and generates data where the two presentations occur independently, which is equivalent to adjusting for HbS in the regression model. The procedure generates simulated data dependent on a parameter characterising the effect of G6PDd on severe malarial anaemia. As no adjustment is necessary, we then compute the non-parametric odds ratio for G6PDd in CM cases versus controls, excluding from the CM case definition all those who have SMA. This simulated 'observed' odds ratio estimated from the 2 x 2 table (cases and controls versus G6PDd and G6PD normals) is thereby directly comparable to the reported odds ratios (obtained from logistic regression with the appropriate adjustments) in *Malaria Genomic Epidemiology Network et al. (2014)*; *Clarke et al. (2017)*, if we assume that only sex, ethnicity and *HbS* are the true confounders (i.e. all necessary adjustments were made in both publications).

The hypothetical data were simulated based on the following assumptions:

1. G6PD deficiency increases the risk of SMA in acute symptomatic malaria (the size of the effect is the only free parameter in the model and varies from 1 to 2 as defined by the odds ratio for G6PDd in SMA cases versus controls). In reality this would be expected to be a function of genotype and gene dose (i.e. hemizygotes and homozygotes would have a greater risk than heterozygotes).
2. CM is independent of G6PD status.
3. A population of males only (i.e. no heterozygote women, so no partial effects) with homogeneous background frequency of G6PDd.
4. CM and SMA occur independently.

From the data in *Clarke et al. (2017)* and *Malaria Genomic Epidemiology Network et al. (2014)*, we can estimate the marginal probability of CM as 0.34, independent of G6PD status (assumption 2). We can also estimate the marginal probability of SMA as 0.24. The probability of G6PDd in males was 0.15.

If we denote $\pi = \mathrm{P(SMA|G6PDd)} \geq \mathrm{P(SMA)} = 0.24$, then by the law of total probability:

$$\mathrm{P(SMA|G6PDn)} = \frac{\mathrm{P(SMA)} - \mathrm{P(SMA|G6PDd)P(G6PDd)}}{1 - \mathrm{P(G6PDd)}} = \frac{0.24 - 0.15\pi}{0.85}$$

where G6PDd denotes G6PD deficient and G6PDn denotes G6PD normal.

The true proportion of G6PDd in the controls is known and fixed as $\mathrm{P(G6PDd)}$, with the proportion of G6PDn in the controls is given by $1 - \mathrm{P(G6PDd)}$, therefore the odds of G6PDd in the control group is given by $\frac{\mathrm{P(G6PDd)}}{1-\mathrm{P(G6PDd)}}$.

We then simulate cases as follows. For each value of $\pi \in [0.24, 0.5]$:

**Step 1**. Simulate 1 million patients such that:

$\mathrm{P(CM)} = 0.34$,

$\mathrm{P(G6PDd)} = 0.15$.

$\mathrm{P(SMA|G6PDd)} = \pi$.

**Step 2**. Select only the patients who have either just CM, or just SMA, filtering out those with concomitant SMA and CM.

**Step 3**. In the remaining data compute $\mathrm{N_{G6PDd}^{SMA}}$ (number of G6PDd with SMA); $\mathrm{N_{G6PDn}^{SMA}}$ (number of G6PDn patients with SMA); $\mathrm{N_{G6PDd}^{CM}}$ (number of G6PDd patients with CM); $\mathrm{N_{G6PDn}^{CM}}$ (number of G6PDn patients with CM). The simulated odds ratio for G6PDd in SMA cases versus controls is $\frac{\mathrm{N_{G6PDd}^{SMA}/N_{G6PDn}^{SMA}}}{\mathrm{P(G6PDd)/(1-P(G6PDd))}}$, and the simulated odds ratio for G6PDd in CM cases versus controls is $\frac{\mathrm{N_{G6PDd}^{CM}/N_{G6PDn}^{CM}}}{\mathrm{P(G6PDd)/(1-P(G6PDd))}}$.

The causal diagram which corresponds to exclusion operating in Step two is shown in *Figure 1*. Selection bias can be seen via the role of the vertex *Case definition* which is a collider between *G6PD deficiency* and *Cerebral malaria (CM)*. Implementation of the simulation model in R can be found at: https://github.com/Stije/SevereMalariaAnalysis/SelectionBiasSimulation.Rmd (*Watson and Leopold, 2019*; copy archived at https://github.com/elifesciences-publications/SevereMalariaAnalysis).

## Additional information

### Funding

| Funder | Grant reference number | Author |
|---|---|---|
| Wellcome Trust | | James A Watson<br>Nicholas PJ Day<br>Arjen M Dondorp |
| National Health and Medical Research Council | Senior Research Fellowship 1104975 | Julie A Simpson |

The funders had no role in study design, data collection and interpretation, or the decision to submit the work for publication.

### Author contributions

James A Watson, Conceptualization, Software, Formal analysis, Validation, Investigation, Visualization, Methodology, Writing—original draft, Project administration, Writing—review and editing; Stije J Leopold, Conceptualization, Data curation, Formal analysis, Validation, Investigation, Visualization, Methodology, Writing—original draft, Project administration, Writing—review and editing; Julie A Simpson, Supervision, Methodology, Writing—review and editing; Nicholas PJ Day, Resources, Funding acquisition, Validation, Writing—review and editing; Arjen M Dondorp, Resources, Supervision, Validation, Writing—review and editing; Nicholas J White, Conceptualization, Resources, Supervision, Funding acquisition, Validation, Writing—review and editing

### Author ORCIDs

James A Watson https://orcid.org/0000-0001-5524-0325
Stije J Leopold https://orcid.org/0000-0002-0482-5689
Julie A Simpson https://orcid.org/0000-0002-2660-2013
Arjen M Dondorp https://orcid.org/0000-0001-5190-2395
Nicholas J White https://orcid.org/0000-0002-1897-1978

### Decision letter and Author response
Decision letter https://doi.org/10.7554/eLife.43154.006
Author response https://doi.org/10.7554/eLife.43154.007

## Additional files

### Supplementary files
• Transparent reporting form
DOI: https://doi.org/10.7554/eLife.43154.004

### Data availability
This manuscript is a methodology paper; no new data were generated. The code for the simulation study can be found on the github repository at https://github.com/Stije/SevereMalariaAnalysis (copy archived at https://github.com/elifesciences-publications/SevereMalariaAnalysis).

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
