## [Decision Letter]

[Editors’ note: this article was originally rejected after discussions between the reviewers, but the authors were invited to resubmit after an appeal against the decision.]

Thank you for submitting your work entitled "Causal pathways in severe falciparum malaria" for consideration by *eLife*. Your article has been reviewed by two peer reviewers, one of whom is a member of our Board of Reviewing Editors, and the evaluation has been overseen by a Senior Editor. The following individual involved in review of your submission has agreed to reveal their identity: Ellie Murray (Reviewer #2).

Our decision has been reached after consultation between the reviewers. Based on these discussions and the individual reviews below, we regret to inform you that your work will not be considered further for publication in *eLife*.

This is a very interesting and thoughtful effort to bring causal reasoning to the extremely complex world of malaria clinical outcomes. Such approaches will be important to generating hypotheses for how to improve clinical care in this field, as the authors note, and I am glad to see these efforts. Toward that end, however, we have substantial concerns about the way it has been applied.

However for reasons stated below, both reviewers have major concerns with the analysis. As a result, this paper cannot be published in something like its current form. The first Case Study is not appropriate because it doesn't ask well-posed causal questions, and the second Case Study is essentially a critique of another paper that I can't verify includes a correct description of what the other paper did. If it is a correct description, it is a very important point that should be made as a letter or some other way, and the simulation is superfluous except to provide an illustration to those outside the epidemiology field.

Normally when *eLife* declines a paper the reviews are appended verbatim to the review, while only revise and resubmit decisions have a combined letter detailing required changes. In this instance, the reviewers had considerable dialogue post review and pre-decision, so we have written a combined letter that contains our key points from the original reviews, revised after discussion.

Major comments:

1) In Case Study I, most of the exposures studied are health states, rather than interventions. As has been discussed in the context of obesity (a health state), the causal question "what is the effect of (a particular health state) on (a particular outcome)" is incoherent, or at best incomplete; causal questions imply a specified (hypothetical) intervention that could be applied in a hypothetical randomized trial to change the health state in a particular way. If the intervention is not specified, then two hypothetical trials employing different interventions to achieve the same change in a patient's health state would potentially get different (even different signs of) results.

The most important implications of a lack of well-defined interventions are (1) that confounding can be hard to define when the way in which a variable is changed is unknown; (2) the practical utility of the estimate is limited since we do not know how to act upon it; and (3) when there are methods of changing the variable that have effects in different directions we are estimating a weighted average and may make the wrong conclusion. For an example of this 3rd issue, consider the exposure weight loss. Some individuals lose weight by diet and/or exercise and this is expected to improve their health status, but other individuals lose weight by developing cancer and this is expected to worsen their health status. If we don't know the relative frequency of these interventions then we might find that losing weight is harmful to overall health even though intentional weight loss via diet and exercise would be beneficial.

To take a key example for this paper, anemia on admission could be changed by (1) moving up the date of admission, presumably to an earlier time in the disease course – e.g. by active case finding and referral; (2) administering blood transfusions on admission; (3) administering drugs to stimulate red cell production on admission; (4) iron dietary supplements at the population level (pre-admission). Even if these were calibrated to achieve exactly the same mean change in hematocrit, (1) could be beneficial; (2) could be slightly harmful (as suggested by the slight increase in mortality in this study); (3) and (4) might go either way, depending on whether they protect the host more or enhance parasite replication more. Similarly, almost any of the other harmful states could be changed on admission by either earlier admission or changing treatment at admission, with potentially different effects. As a result, even if all confounding pathways were blocked, the interpretation of the resulting estimate would be as a weighted average of the effects of 1-4 (and any other possible methods of changing anemia) with unknown weights, making the answer uninterpretable for clinical or public health decision-making.

By this logic, the effects in Figure 4, with the exception of the treatment effect, are not the answers to well-posed causal questions and should be removed.

We note that both reviewers, who are knowledgeable about causal inference, acknowledge that the view taken here – that well-defined interventions are required for a well-posed causal question – is not universal in the field. Some have defended the use, for example, of race or sex as causes of disease outcomes, without a well-defined notion of how an intervention could change these. However, we make a distinction between effectively "unalterable" states (such as sex and race), about which controversy exists as to whether they can be considered in causal analyses, and readily alterable states (such as hematocrit) in which very straightforward alterations (transfusion, nutrition, earlier presentation) plausibly have opposite signs of effect. Here, using the machinery of causal inference to isolate the "causal" effect of hematocrit does not helpfully inform interventions, because the interventions might have the same or the opposite effect as suggested by the analysis. Besides this there were other concerns about the causal reasoning in Case Study I.

2) It was not clear that the DAG in Figure 3 supports the authors conclusions about the identifiability of a causal effect of parasitemia as drawn – in the fifth paragraph of the subsection Case study 1: factors determining survival in "severe malaria", this is described as biased through an open path at immunity, while in the subsection "Simulation study for case study 2" as through an open path at anaemia. But, assuming that all the blue and green nodes represented measured covariates which can be adjusted for, there are no open backdoor paths from parasitemia to death through either immunity (the path becomes blocked again at anaemia) or anaemia itself (the open collider path is blocked at age). However, it is certainly reasonable that there exist other unmeasured common causes of anaemia or immunity (for example, the gene from case study 2 is not included in this DAG, nor are diet or other co-infections), and if those were included (or if some of the green or blue nodes especially age or AKI were not measured) there would indeed be an open backdoor path. Most common biases in real data analyses occur because of paths that involve unmeasured variables, so including unmeasured variables such as genes, diet, or coinfections as unknowns in your figure would also be useful for didactic purposes. Furthermore, it's not clear that it would be appropriate to condition on all the blue and green variables for exactly this reason – they are likely to be colliders and therefore conditioning can induce bias; and conditioning on them would remove some of the effect of parasitemia.

3) If this paper is intended to be a tutorial for readers who have not had prior exposure to causal inference or directed acyclic graphs, the description of the assumptions required for these methods and their comparison to other approaches is insufficient. A simpler example would be more useful and the paper should probably include more detail on how to read the DAG to determine potential for bias. In general, a separate DAG should be drawn for each causal question – the DAG in Figure 3 contains many more variables than are necessary to estimate an effect of parasitemia (presuming we could define one sufficiently well), but does not include sufficient variables to estimate the effect of anti-malarial drugs except from a randomized controlled trial (and in that case has too many variables). Finally, it seems likely that a number of these variables have a cyclic / feedback relationship, in which case there may be time-varying exposure-confounder feedback which could exacerbate the problems of a lack of well-defined interventions.

4) For Case Study II, it is hard to understand what was done in Clarke et al., the paper in *eLife* that they criticize. We can't tell on a brief look at that paper whether the comparison was between:

a) Severe malaria anemia (SMA) (+- cerebral malaria CM) vs. population controls, and separately CM+-SMA vs. population controls, for G6PD status (which would not suffer from the problem the authors posit);

b) SMA only vs. population controls, and CM only vs. population controls (in which the CM only group would be depleted of those with SMA, and thus a risk factor for SMA would falsely look protective against CM only);

c) SMA vs. CM among severe malaria (which appears to be the case for the DAG presented in Figure 7, though I'm not sure), which seems to be what the R code posted on GitHub assumes.

This needs to be clarified further before it can be evaluated.

[Editors’ note: what now follows is the decision letter after the authors submitted an appeal.]

I and the reviewers have read and considered your letter of appeal, which makes a number of thoughtful points. After discussion among the reviewers, our view is as follows:

1) The first half of the paper could be a focused effort to assess the evidence for the effectiveness of rapid transfusion in preventing malaria mortality, with causal machinery used to improve the credibility of this inference compared to more informal methods. Such a study would avoid attributing causal effects to state variables such as hematocrit or temperature or the like, and would focus on a well-defined intervention. Without resolving the philosophical question of whether causal questions can ever be asked without a well-defined intervention, it seems that in the context of acute malaria mortality, with an explicit justification in the introduction that the authors seek to inspire clinical studies, this is a reasonable constraint. If you resubmit a version with this issue, please also attend to the other issues raised by the second reviewer.

2) The second half of the paper is really quite unrelated, except by the use of causal methodology, and while it seems to make a very important point, it does so in a way that is confusing in the sense that the back-story and evidence for what was done in previous studies is obscure. In my personal opinion, this seems to be a short paper of its own, where the history of prior studies needs to be presented in such a way that the reader can clearly tell what has been done, before the authors of the present manuscript explain why that is fallacious.

3) There is a third, implicit purpose of the paper, which is to introduce the malaria clinical epi community to causal reasoning. This is in our view not ideally done by the first half of the paper, which is extremely complex and raises the issue of what is a causal question as well as the rather complicated mechanics of how to control for various confounders. The second half may be better suited to this purpose as it is simpler.

We can think of a few different ways you might proceed; a focused paper on the second part (which would seem appropriate for *eLife* given the publication of the prior paper in the journal); a tutorial paper that uses the second part as an example but is more generally about causal reasoning; or perhaps the full paper (in which case we would suggest switching the order of the parts because the G6PD piece is easier to understand so could naturally come first). This last possibility seems unwieldy for the same reasons as the original paper.

You are free to take or leave any of these suggestions, but this is how we see the paper. If you do split it up, we would suggest that the G6PD part (with or without surrounding tutorial material) would be of greatest interest to *eLife* given the prior publication. The mortality prediction part is a large and complex approach to a focused clinical question that might by itself be a very good paper for a more specialized journal.

[Editors’ note: what now follows is the decision letter after the authors submitted for further consideration.]

Thank you for submitting your article "Berkson's bias and the apparent protective effect of glucose-6-phosphate dehydrogenase deficiency on cerebral malaria" for consideration by *eLife*. Your article has been reviewed by two peer reviewers, and the evaluation has been overseen by a Reviewing Editor and Neil Ferguson as the Senior Editor. The reviewers have opted to remain anonymous.

The reviewers have discussed the reviews with one another and the Reviewing Editor has drafted this decision to help you prepare a revised submission.

This paper attributes findings of a protective effect of G6PD deficiency against cerebral malaria to a statistical/causal artifact known as Berkson's bias (though maybe not the orthodox version thereof) or more generally selection or collider bias. It uses argumentation and simulation to suggest that the problem is:

1)G6PDd is a cause of increased risk of severe malarial anemia (SMA);

2) Two prior studies compared cases with various forms of severe malaria, specifically SMA and CM, to community controls for the prevalence of G6PDd, and found it was higher than controls in SMA and lower than controls for CM. They concluded a likely causal protective effect of G6PDd on CM;

3) The case definitions used were CM alone (i.e. no other severe malaria diagnosis) and SMA alone (i.e. no other severe malaria diagnosis. Under the assumption that CM and SMA risk are independent complications of malaria, and that G6PDd predisposes to SMA, this exclusion means that the CM cases in the analysis were SMA-free, thus less likely to have a predisposing allele to SMA. This would show up as a "protective" effect of the allele on CM because CM really means, in this study, CM and not SMA;

4) Simulation shows this could account for the full effect.

The reviewers and reviewing editor have discussed the manuscript and believe it can be made acceptable for publication after some crucial but relatively minor revisions:

1) Clarity about what was done. Reviewers had a hard time establishing precisely what was done in the two papers, and how that relates to the simulations. In particular, please:

a) State precisely the regression that was made in each paper [which we believe was Pr(CM and not SMA) vs. Pr (control), in logistic regression with G6PD genotype as predictor in a case-control format, and similarly for SMA and not CM) and quote the relevant passage in each paper that states the exclusion of dual cases.

b) State that this is the same comparison (with certain simplifications e.g. males only) in the simulation.

2) Address a reviewer concern that the statistical noise around the estimated OR looks large in your simulations given the use of 1 million people. This may be a false impression or may be due to the rarity of one cell in the odds ratio, but please explain.

3) Address a reviewer concern that producing a close quantitative match to the biased odds ratio for CM is not easily interpretable given the simplifying assumptions notably males only – would that not change the value substantially so that the agreement becomes qualitative rather than quantitative? A simulation including females would be simple to do.

4) Reviewers found point #1 confusing perhaps for several reasons, one of which is the use of Berkson's bias as the explanation here. Indeed, on first reading I had thought that the mistake was that the earlier papers had looked at predictors of SMA and CM among all severe malaria patients (without healthy controls). That would be classic Berkson's bias as taught in basic epidemiology classes. The bias you have identified is closely related, is still a form of collider bias, but is not exactly the same; it is that "CM" is really "CM and not SMA" and vice versa. The wording about Berkson's bias may just mislead people – maybe you want to say collider bias, and make clearer in the DAG how this works. Removing Berkson from the title could also clarify for those who only read the title!

---

## [Author Response]

[Editors’ note: the author responses to the first round of peer review follow.]

[…] However for reasons stated below, both reviewers have major concerns with the analysis. As a result, this paper cannot be published in something like its current form. The first Case Study is not appropriate because it doesn't ask well-posed causal questions, and the second Case Study is essentially a critique of another paper that I can't verify includes a correct description of what the other paper did. If it is a correct description, it is a very important point that should be made as a letter or some other way, and the simulation is superfluous except to provide an illustration to those outside the epidemiology field.

The summary letter clearly demonstrates that both the editors and the reviewers have spent considerable time delving into the paper and the accompanying code. We are very grateful. The synthesized criticisms are informed and constructive and this assessment and review is much appreciated and will ultimately lead to a substantially improved manuscript.

We would like to appeal against the decision to reject the paper.

The major concern (point 1) is that Case Study 1 does not ask well-posed causal questions because it does not discuss interventions. However, both reviewers acknowledge that this view is not universal in the field of causal inference and that this perspective is subject to some debate. We accept that consideration and presentation of this important aspect was lacking in the submitted version of the paper. However, contextual understanding of the medical aspects of severe malaria is needed for a correct interpretation of our results. Severe malaria is a medical emergency with high mortality, and a very rapid evolution. Most deaths occur in the first 24 hours following admission to hospital resulting from sudden onset of acute complications. The pathological process and therapeutic implications are more comparable to haemorrhage. There is a single well-defined and feasible intervention: blood transfusion. Indeed, severe malaria is the major reason for blood transfusion in children in sub-Saharan Africa. Severe malaria is the consequence of the malaria parasite invading a substantial proportion of circulating red blood cells, and these invaded red cells blocking the microcirculation – a process with a time course measured in hours. Anaemia results largely from obligatory haemolysis following schizogony, and sequestration of infected and uninfected red cells. To this extent, the primary way in which the variable haematocrit changes is clear. Furthermore, the degree of anaemia (measured as haematocrit) is certainly a determinant of patient outcome and has considerable practical utility. The counterfactual outcomes of interest are those that would be observed if it were possible to change the haematocrit on admission. Transfusion on admission is the only acute intervention to treat severe malaria in this emergency situation. The other interventions that are mentioned to treat anaemia are not relevant in this context. Patients would die before any drugs acting on the blood marrow took effect; active case detection cannot be done in remote rural areas where severe malaria kills most children; iron supplements at a population level are not a reasonable intervention providing no protection against acute haemolysis causing anaemia in falciparum malaria. Thus, in the case of haematocrit, the way in which this variable changes is known; there is practical utility in its measurement; and the only relevant intervention is transfusion. For these reasons we do not think that the measured causal effect can be explained as a weighted average of multiple factors. However, we agree with the reviewers, that this might not the case for each of the other variables.

The other critique is the dependence on the time of admission. Every study in severe malaria suffers from the selection bias of only considering patients who seek treatment at larger health centres. However, the problem addressed in our paper addresses the real-life situation on how to manage patients with severe falciparum malaria admitted to hospital. Whether or not to transfuse these patients on admission is a very important question and we believe causal reasoning is essential in the debate. We find that moderate anaemia is not harmful and could be even be beneficial in severe malaria. We are careful not to overstate the implications for blood transfusion due to the exploratory nature of the analysis. Our finding may explain why the very large FEAST study reported a six-fold higher mortality in patients who were above the recommended haematocrit transfusion threshold yet still received a blood transfusion. In summary, the causal question for haematocrit is well-posed and this work should help guide future analyses of transfusion-based intervention studies.

Well-defined interventions are also potentially available for raised blood urea nitrogen: the intervention is here is hemofiltration or dialysis; high parasitaemia: the intervention is the anti-parasitic drug; pulmonary oedema: the intervention is oxygen and positive pressure ventilation. For seizures, anticonvulsants can be given. For acidosis there are experimental treatments. For coma itself there are no direct interventions possible and we agree to remove this from Figure 4. However, we think anaemia, pulmonary oedema, seizures and blood urea nitrogen should stay in Figure 4. None of these variables are chronic unalterable health states but the result of the acute infection.

The second major concern (point 4) is that Case Study 2 is a critique of two recent major papers (notably one in eLife) and that the validity of this critique cannot be verified. Both Nature genetics and eLife only accept correspondence up to one year after publication (publication dates were 2014 and January 2017, respectively), and therefore this channel of communication is closed. The availability of open channels for balanced critiques of published research is essential for reproducible research. We are sure that eLife promotes this healthy scientific exchange. We have asked Professor Kwiatkowski (corresponding author for both papers) which of the possible analyses were done and it was confirmed to be scenario b (from your summary letter). When we subsequently sent our concerns along with a draft manuscript (in total two emails), no response was given. We have the email correspondence to verify these exchanges.

This highlights the problems of publishing data analytic results with no accompanying code. Even if their data were openly available, reproducing the analysis would require reverse engineering. We have gone to significant effort to support all our results with open, carefully annotated code and deidentified data.

As a final point, it is mentioned that the simulation study we present in Case Study 2 is superfluous. We respectfully disagree. This simple simulation shows that reasonable values of the effect of G6PD deficiency on anaemia (taken from previous published estimates) would suffice to explain the observed negative association between cerebral malaria and severe malarial anaemia. This `sensitivity’ analysis is an important building block in the argument: if larger, unreasonable effects were necessary to explain the observed results, a residual direct effect could still be posited.

Would eLife consider our appeal under the following set of conditions?

· We will add to the Methods section careful consideration of the counterfactual states underlying the exposure variables.

· We will remove coma as an exposure with an interpretable causal effect and add a point to this effect in the discussion.

· We will address the concerns in point 2. This highlights an error in the manuscript and this will be changed accordingly.

· We disagree with point 3. All variables considered are at the point of admission and therefore there are no feedback loops involved. The didactic aspect will also be addressed, but the DAG in Figure 3 is useful in that it shows the overall picture of expert knowledge in severe malaria using the key admission variables.

[Editors’ note: the author responses to the re-review follow.]

[…] We can think of a few different ways you might proceed; a focused paper on the second part (which would seem appropriate for eLife given the publication of the prior paper in the journal); a tutorial paper that uses the second part as an example but is more generally about causal reasoning; or perhaps the full paper (in which case we would suggest switching the order of the parts because the G6PD piece is easier to understand so could naturally come first). This last possibility seems unwieldy for the same reasons as the original paper.You are free to take or leave any of these suggestions, but this is how we see the paper. If you do split it up, we would suggest that the G6PD part (with or without surrounding tutorial material) would be of greatest interest to eLife given the prior publication. The mortality prediction part is a large and complex approach to a focused clinical question that might by itself be a very good paper for a more specialized journal.

This is a re-submission of a previously rejected paper, entitled "Causal pathways in severe malaria". Following the recommendations of the senior editor (Prof. Neil Ferguson), after a successful appeal against the reject decision, we have made considerable changes to the manuscript. We have kept only the section which deals with Berksonian bias in two recent major publications (Malaria Gen Nature Genetics 2014, and Clarke et al., 2017) which report a protective effect of G6PD deficiency in severe cerebral falciparum malaria. Our re-submission focuses on highlighting how Berkson’s bias likely explains all of the observed association. This would invalidate the model of balancing selection for G6PD deficiency mutations proposed by Clarke et al.

The only relevant reviewer comment concerning this re-submission is now:

4) For Case Study II, it is hard to understand what was done in Clarke et al., the paper in eLife that they criticize. We can't tell on a brief look at that paper whether the comparison was between:a) Severe malaria anemia (SMA) (+- cerebral malaria CM) vs. population controls, and separately CM+-SMA vs. population controls, for G6PD status (which would not suffer from the problem the authors posit);b) SMA only vs. population controls, and CM only vs. population controls (in which the CM only group would be depleted of those with SMA, and thus a risk factor for SMA would falsely look protective against CM only);c) SMA vs. CM among severe malaria (which appears to be the case for the DAG presented in Figure 7, though I'm not sure), which seems to be what the R code posted on GitHub assumes.This needs to be clarified further before it can be evaluated.

The Materials and methods section of the paper now shows that scenario 3 is almost certainly the one used in Clarke et al.to demonstrate a protective effect of G6PD deficiency on cerebral malaria. Although it is not directly stated in the Clarke et al. Materials and methods section, we show that it can be deduced from tabular data presented in the paper.

We suggest that the article type of this re-submission is modified from `Research Article’ to `Short Report’ in light of its new brevity.

[Editors’ note: the author responses to the re-review follow.]

The reviewers and reviewing editor have discussed the manuscript and believe it can be made acceptable for publication after some crucial but relatively minor revisions:1) Clarity about what was done. Reviewers had a hard time establishing precisely what was done in the two papers, and how that relates to the simulations. In particular, please:a) State precisely the regression that was made in each paper [which we believe was Pr(CM and not SMA) vs Pr (control), in logistic regression with G6PD genotype as predictor in a case-control format, and similarly for SMA and not CM) and quote the relevant passage in each paper that states the exclusion of dual cases.b) State that this is the same comparison (with certain simplifications e.g. males only) in the simulation.

We agree that the original presentation lacked clarity and we have substantially rewritten the Materials and methods section in light of this comment (subsection "Data analysis in Clarke et al., 2017, and MalariaGEN et al., 2014".

The rationale for why we believe our simulation mimics these reported results is then given in the subsection "Sensitivity analysis".

The bias stems directly from the selection distortion in the CM case definition. The adjustment for sickle cell in the logistic regressions in both MalariaGen, 2014, and Clarke et al., 2017, will not impact on the bias we are investigating in our simulation study. Sickle cell is a confounder between the two clinical presentations – as demonstrated in the causal diagram (Author response image 1), an extension of Figure 1 in our paper to include a vertex for sickle cell status – and the authors are correct to adjust for it, therefore removing any biased association. Our simulation assumes that SMA and CM occur independently thereby reflecting the adjusted relationships in both papers (subsection "Sensitivity analysis").

**Author response image 1. respfig1:** Role of sickle cell in case definitions of SMA and CM. Sickle cell mutations will increase the likelihood of anaemia and are presumed protective against CM.

2) Address a reviewer concern that the statistical noise around the estimated OR looks large in your simulations given the use of 1 million people. This may be a false impression or may be due to the rarity of one cell in the odds ratio, but please explain.

This was a typo in the code – the original plot was not based on 10^5^ rather than 10^6^ individuals. We have corrected this and the noise in the plot (Figure 2) is greatly reduced. We thank the reviewer for pointing this out!

3) Address a reviewer concern that producing a close quantitative match to the biased odds ratio for CM is not easily interpretable given the simplifying assumptions notably males only – would that not change the value substantially so that the agreement becomes qualitative rather than quantitative? A simulation including females would be simple to do.

In fact, we contrast our simulated results against the reported results in males only. We get an almost perfect match for the reported results in males. This has been made clearer in the Results (second paragraph). Therefore, in our opinion, there are no major simplifying assumptions that should impact this simulation/sensitivity analysis.

Although it is simple to generate proportions of females and males with G6PDd under an assumption of Hardy-Weinberg equilibrium, a simulation involving females is in fact more complicated. It necessitates assumptions regarding the gene dose effect (female heterozygotes are mosaics of deficient and normal red blood cells). As the simulation is currently written, there is a unique free parameter and this facilitates interpretation. The gene dose effect would require one extra parameter for which there are little data on which to calibrate it.

4) Reviewers found point #1 confusing perhaps for several reasons, one of which is the use of Berkson's bias as the explanation here. Indeed, on first reading I had thought that the mistake was that the earlier papers had looked at predictors of SMA and CM among all severe malaria patients (without healthy controls). That would be classic Berkson's bias as taught in basic epidemiology classes. The bias you have identified is closely related, is still a form of collider bias, but is not exactly the same; it is that "CM" is really "CM and not SMA" and vice versa. The wording about Berkson's bias may just mislead people – maybe you want to say collider bias, and make clearer in the DAG how this works. Removing Berkson from the title could also clarify for those who only read the title!

We agree and thank you for this suggestion. The title has been changed to "collider bias" and we have changed all mentions in the main text to collider bias also.

In the DAG in Figure 1, we have changed the name of the collider variable from "Included in study" to "Case definition". This shows how the case definition is simultaneously dependent on both clinical presentations.